# Influence of nutrient status on the response of the diatom *Phaeodactylum tricornutum* to oil and dispersant

Manoj Kamalanathan[1]*, Jessica Hillhouse[1], Noah Claflin[1], Talia Rodkey[1¤a], Andrew Mondragon[1], Alexandra Prouse[1], Michelle Nguyen[1¤b], Antonietta Quigg[1,2]

**1** Department of Marine Biology, Texas A&M University at Galveston, Galveston, Texas, United States of America, **2** Department of Oceanography, Texas A&M University, College Station, Texas, United States of America

¤a Current address: Lehigh University in Bethlehem, Bethlehem, Pennsylvania, United States of America
¤b Current address: College of Earth, Ocean, and Atmospheric Sciences, Oregon State University, Corvallis, Oregon, United States of America

* manojka@tamug.edu, manojkamalanathan711@gmail.com

## Abstract

Phytoplankton play a central role in our ecosystems, they are responsible for nearly 50 percent of the global primary productivity and major drivers of macro-elemental cycles in the ocean. Phytoplankton are constantly subjected to stressors, some natural such as nutrient limitation and some manmade such as oil spills. With increasing oil exploration activities in coastal zones in the Gulf of Mexico and elsewhere, an oil spill during nutrient-limited conditions for phytoplankton growth is highly likely. We performed a multifactorial study exposing the diatom *Phaeodactylum tricornutum* (UTEX 646) to oil and/or dispersants under nitrogen and silica limitation as well as co-limitation of both nutrients. Our study found that treatments with nitrogen limitation (-N and–N-Si) showed overall lower growth and chlorophyll *a*, lower photosynthetic antennae size, lower maximum photosynthetic efficiency, lower protein in exopolymeric substance (EPS), but higher connectivity between photosystems compared to non-nitrogen limited treatments (-Si and +N+Si) in almost all the conditions with oil and/or dispersants. However, certain combinations of nutrient limitation and oil and/or dispersant differed from this trend indicating strong interactive effects. When analyzed for significant interactive effects, the–N treatment impact on cellular growth in oil and oil plus dispersant conditions; and oil and oil plus dispersant conditions on cellular growth in–N-Si and–N treatments were found to be significant. Overall, we demonstrate that nitrogen limitation can affect the oil resistant trait of *P. tricornutum*, and oil with and without dispersants can have interactive effects with nutrient limitation on this diatom.

## 1. Introduction

Phytoplankton, the driver of elemental cycles in the ocean (Carbon, Nitrogen, Phosphorus, Sulphur, Silica etc.), are integral to our ecosystems [1–3]. Changes to their composition,

**Data Availability Statement:** Gulf of Mexico Research Initiative Information and Data Cooperative at DOI: 10.7266/1TBZJDET.

**Funding:** This study was funded by the Gulf of Mexico Research Initiative (grant number ADDOMEX2) awarded to AQ. This study was also funded by the NSF REU Program (grant number 1560242) awarded to TR. The funders had no role in study design, data collection and analysis, decision to publish, or preparation of the manuscript.

**Competing interests:** NO authors have competing interests.

growth and physiology can have a wide array of cascading impacts through the oceanic food-webs [4]. This is particularly the case during natural and man-made environmental disasters such as hurricanes and oil-spills [5–8]. Nitrogen (N) required for protein synthesis (hence enzymes) is essential for growth, photosynthesis, and survival, yet many parts of the global oceans and coasts are nitrogen-limited for phytoplankton growth [9]. In addition to N required for proteins involved in phytoplankton central metabolism, membranes and nucleic acid synthesis, N is also needed for additional critical reactions such as carbon fixation (RUBISCO), light harvesting proteins, and chlorophyll biosynthesis. Therefore, N is a critical component of photosynthetic machinery and for cellular maintenance [10] and its limitation can have cascading top-down effects on the amount of carbon and energy acquisition and their transport [11]. Laboratory studies show that phytoplankton undergo physiological changes to allow for growth and photosynthesis under a lower cellular N budget [12–15]. While all phytoplankton need N and other macro-elements, diatoms (Bacillariophycease) are unique in also having a requirement for silicate (Si) to build their frustles [16].

The Gulf of Mexico is also home to one of the largest oil and gas exploration zones, which makes the site more susceptible to man-made disasters such as oil-spills. According to NOAA incident news [17], there have been over 11 oil spills in this region since 2019 alone, out of the > 180 national oil spills. If an oil spill coincides during a period when phytoplankton are typically N limited, it can potentially become an additional stressor, but the effects of the combination requires further investigation. Phytoplankton in the Gulf of Mexico are found to be frequently N limited, but also can be phosphorus or light limited [18–20], with N limitation most likely to occur in high salinities/summer time [21]. A recent study found that if the N limitation is alleviated, the microbial community responds in a variety of ways including enhanced oil degradation and changes in community composition [22, 23]. Moreover, oil exposures are more toxic to nitrifiers, as opposed to most denitrifiers, which can also drive loss of nitrogen though denitrification, thereby affecting phytoplankton [24]. In addition, dispersants are an important tool for remediation of oil spills, however, several studies have shown negative effects of dispersants on the growth of phytoplankton, thereby adding another layer of stress [6, 25]. One of the objectives of this study is to understand the effects of combinations of N limitation, oil and dispersant exposure.

Previous studies by Bretherton et al., (2018, 2020) have shown that phytoplankton have a spectrum of responses to oil exposure, with some being sensitive and some tolerant [26, 27]. Of the cell traits examined, cell size was found to be most important in determining the bio-mass response to oil, whereas motility/mixotrophy was more important in the dispersed oil among the 15 species examined. Further, Bopp et al. (2007) and Carvalho et al., (2011a, b, & c) hypothesized that exposure to polycyclic aromatic hydrocarbons (PAHs) component of oil can interfere with the silica (Si) transport of the cell and affect the growth of the diatoms during oil exposure [28–31]. This implies that the ability to form silica frustule can play a critical role in survival of diatoms during an oil spill. In addition, Dortch and Whitledge (1992) also suggested that the Gulf of Mexico is more likely to be silicate limited than nitrogen [21]. Therefore, the second objective of this study is to test this hypothesis by using a combination of oil and dispersant exposure with Si limitation. The third objective of this study is to test whether there is a synergy of negative effects between oil and/or dispersant exposure with N and/or Si limitation.

*Phaeodactylum tricornutum* (UTEX 646) is a pennate diatom and widely considered as one of the model species for physiological studies. *P. tricornutum* was chosen for this study for two reasons. First, it has been shown to be resistant to oil and dispersant exposures [26], so our study aims to test the effects of oil, dispersant and N limitation on phytoplankton that are likely to survive an oil-spill. Second, this diatom has the unique ability to grow in the absence of

silica, thereby allowing us to test the hypothesis of the role of silica frustule formation by comparing its response to oil in the presence and absence of Si. By using *P. tricornutum*, we performed a multi-factorial study to understand the outcome of oil and remedial dispersant application on phytoplankton during nitrogen and silica limitation.

## 2. Material and methods

### 2.1 Culture maintenance and experimental design

*P. tricornutum* (Culture Collection of Algae at The University of Texas at Austin–UTEX 646) was maintained at 60 μmol photons $m^{-2} s^{-1}$ and a 12-h/12-h light/dark cycle at 19˚C in Artificial Sea Water (ASW) medium [32]. The study consists of four different experiments with a total of 16 treatments. For treatments 1–4, there are triplicate Control, water accommodated fraction of oil (WAF), chemically enhanced water accommodated fraction of oil (CEWAF), and diluted chemically enhanced water accommodated fraction of oil (DCEWAF). *P. tricornutum* cultures used for this round of experiments were grown under nutrient replete conditions for many (>20) generations. Treatments 5–8 include *P. tricornutum* grown into N limitation combined with treatments 1–4, these treatments are termed as–N, WAF-N, CEWAF-N, and DCEWAF-N. Nitrogen limited cultures used for these experiments were grown in ASW with ¼ of the amount of original N for 2 months and transferred to media with no added N at the start of the experiment. Preliminary experiments with direct transfer of *P. tricornutum* to media without N did not show any effects highlighting the need to incubate in lower nitrogen concentration (in our case¼ of the amount of original N) to achieve nitrogen limitation in our experiments. In addition, the nitrogen limitation status was determined by measuring the maximum quantum yield of photosystem (PS) II ($F_v/F_m$; relative units; see below for details), a reliable indicator of nutrient stress [33]. Treatments 9–12 include *P. tricornutum* grown into Si limitation combined with treatments 1–4, these treatments are termed as–Si, WAF-Si, CEWAF-Si, and DCEWAF-Si. Silica limited cultures used for these experiments were grown in ASW with no added Si for 2 months prior to starting the experiments. Treatments 13–16 include N and Si limitation combined with treatments 1–4, these treatments are termed as–N–Si, WAF–N-Si, CEWAF–N-Si, and DCEWAF–N-Si. Nutrient limited cultures used for these experiments were grown in media with no added Si and ¼ N for a minimum of 5 months and then transferred to media with no added Si or N at the start of the experiment. The longer time periods used for–N–Si nutrient limitation were simply to gain sufficient biomass to conduct the experiments at the same cell densities as the other experiments. +N+Si and +Si labelled treatments indicate that these treatments had full concentration of both nitrogen and silica and just silica in them. All experiments were performed in triplicate in one liter glass bottles with no agitation.

Cell density at the start of experiments was set at $10^5$ cells $mL^{-1}$. WAF, CEWAF and DCEWAF were prepared using the standard CROSERF method [34] with modifications as described in Bretherton et al. (2018) [26]. Briefly, 400 μL of Macondo surrogate oil was added per L of sterile ASW media and stirred overnight in the dark in aspirator bottles. CEWAF was prepared by pre-mixing oil with Corexit in a ratio of 20:1 to be consistent with previous studies [26, 27, 35–37]. A similar process was used to prepare WAF. DCEWAF was prepared by diluting the CEWAF ten-fold with ASW. The media containing oil were filtered using a 20 μm Teflon sieve to obtain WAF, DCEWAF and CEWAF free of large oil droplets. Oil concentrations in the treatments were measured as estimated oil equivalents (EOE) according to Wade et al. (2011) [38]. The average EOE concentrations in WAF, DCEWAF and CEWAF were 2.53 (± 1.93 mg.$L^{-1}$), 13.76 (± 5.24 mg.$L^{-1}$) and 37.16 (± 11.77 mg.$L^{-1}$) respectively. For all the parameters tested in the experiments, the sampling time points were chosen based on the typical

growth curves of *P. tricornutum* to accommodate the initial time point (Day 1), the logarithmic phase (Day 4) and the stationary phase (Day 7) effects.

## 2.2 Growth, morphology and photo-physiology

Growth was monitored by microscopic cell counts on a Neubauer hemocytometer. Morphological observation of features such as cells in chains were made using Imaging FlowCytobot (Mclane labs). Briefly, 5 mL of cells were added to a mixture of 20 mL filtered seawater and 20mL of DI water to dilute the sample. 5 mL of the diluted sample was then run through the Imaging FlowCytobot (Mclane Laboratories), this process was repeated for each treatment.

Photo-physiological parameters measured included chlorophyll *a* ($\mu$g.cell$^{-1}$), maximum quantum yield of photosystem (PS) II ($F_v/F_m$; relative units), light harvesting ability ($\alpha$; $\mu$mol e$^-$. $\mu$mol photons), connectivity of PS II ($\rho$; relative units), $Q_A$ re-oxidation rates ($\tau$; $\mu$sec), and maximum absorption cross-section area ($\sigma$; Å$^2$ quanta$^{-1}$). Chlorophyll *a* was measured using a Turner fluorometer, whereas the other photo-physiological parameters were measured using a Fluorescence Induction and Relaxation Fluorometer System (Satlantic) as per methods described in Bretherton et al. (2018) [26].

## 2.3 Exopolymeric substances (EPS)

Exopolymeric substances were measured by summing the concentration of extracellular neutral sugars, proteins, and uronic acids. Cultures (50 mL) were filtered on a glass-microfiber filter (GF/F) while collecting the filtrate for EPS analysis. The EPS was then concentrated by using an Amicon Ultra-15 centrifugal filter unit with ultracel-3 membrane (Millipore, 3 kDa). The material collected in the 3 kDa filter was used for neutral sugars, protein and uronic acid estimation as per described in Kamalanathan et al., (2018b) [36]. Briefly, neutral sugars was determined by Anthrone method with glucose as the standard [39], and the protein content was determined using the Pierce BCA protein assay kit using bovine serum albumin as the standard [40]. Uronic acids were estimated according to Blumenkrantz and Asboe-Hansen (1973) by the addition of sodium borate (75 mM) in concentrated sulfuric acid and m-hydroxydiphenyl using glucuronic acid as the standard [41].

## 2.4 Logistic regression modelling and interaction analysis

The effect of oil concentrations on relative cellular levels was analyzed using a generalized linear model in R. The cellular concentrations were normalized for all the experiments by calculating the percent change in growth relative to Day 1 for this analysis. EOE values measured across different time points during the experiments across the different treatments and conditions were used as oil concentration for this analysis. The effects of nutrient limitation and the presence of dispersants and their interaction on the relationship between oil concentrations and relative cellular growth were also analyzed using the equation: glm (Relative cellular levels ~ Oil concentration + Treatments x Conditions). The interaction means were calculated and compared using Phia package [42].

## 2.5 Statistics

All treatments were performed in triplicate with findings presented as means plus or minus standard deviation. All parameters excluding the logistic regression modelling were tested for significance using Two-way ANOVA in R using vegan package.

## 3. Results

### 3.1 Growth and morphological response of *P. tricornutum*

The growth response of *P. tricornutum* was determined for all the treatments (–N,—Si, -N-Si, and +N+Si) in all the conditions (Control, WAF, DCEWAF and CEWAF) by calculating the percentage relative increase in cell density on Day 4 and 7 compared to Day 1 (Fig 1A and 1B respectively). This was performed to account for the largely differing initial cell concentration amongst the various conditions and treatments, and to allow for direct comparison of the extent of growth inhibition on a single scale. However, raw growth curves are presented in S1 Fig. When relative growth of *P. tricornutum* at the initial time point was compared, +N+Si overall had higher growth relative to all the other treatments on Day 4 and 7 (Two-way ANOVA; $p < 0.004$), except for–Si treatment, which had significantly higher growth observed in the DCEWAF than the +N+Si (Two-way ANOVA; $p = 0.000059$) on Day 4 (Fig 1A).–N-Si treatment in all the conditions had the lowest overall growth observed compared to all the treatments in all the conditions on Day 4, followed by–N (Fig 1A). On Day 4, a decreasing growth pattern was observed for +N+Si treatment in the following order Control > WAF >

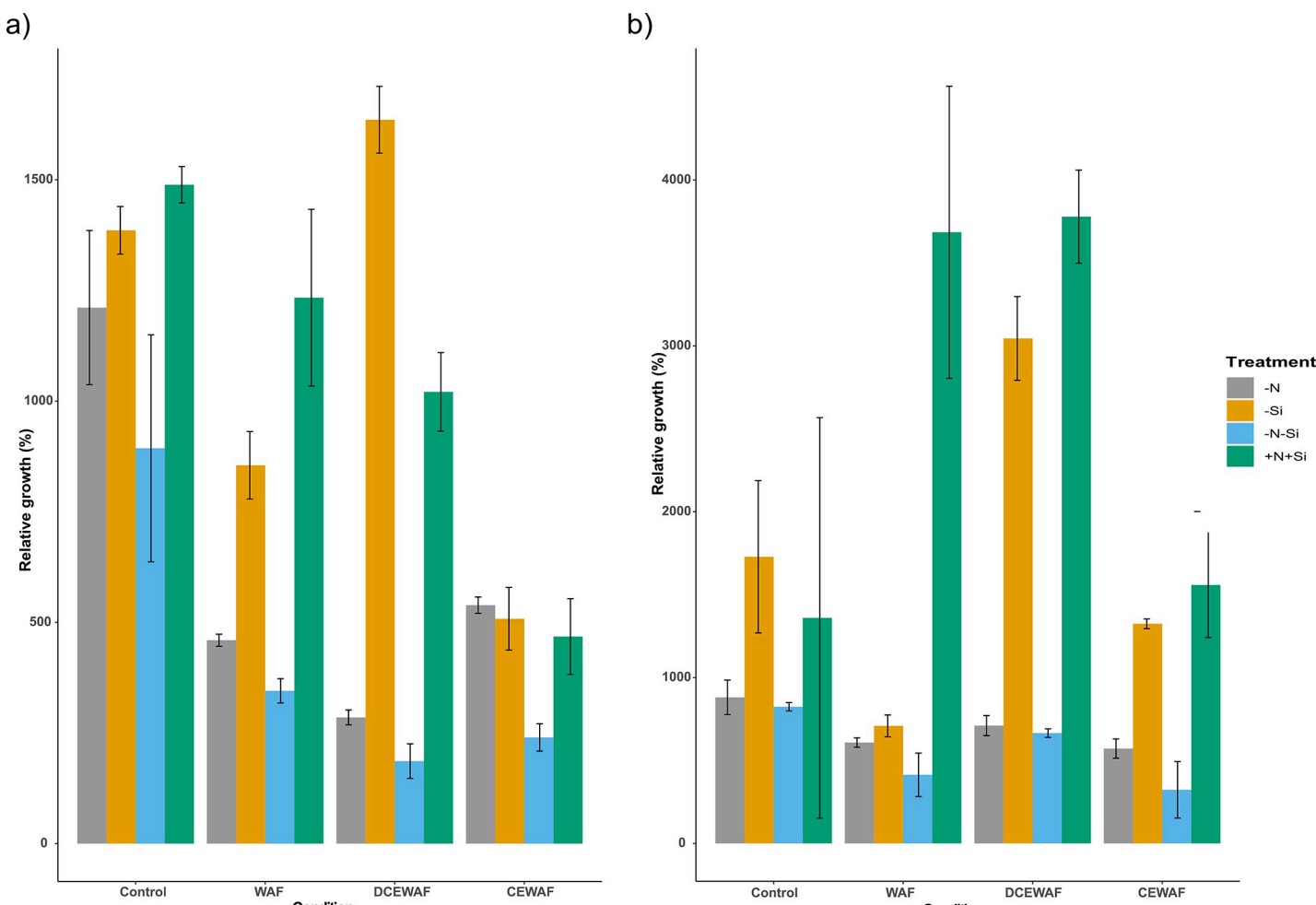

**Fig 1. Growth response of *P. tricornutum*.** a) average growth on Day 4 relative to Day 1 (%), b) average growth on Day 7 relative to Day 1 (%) (± standard deviation) under different treatments and conditions (n = 3). The symbols–N, -Si, -N-Si, and +N+Si indicate nitrogen limited, silica limited, both nitrogen and silica limited and nitrogen and silica replete treatments.

DCEWAF > CEWAF, and a similar pattern was observed for–N and–N-Si treatments but only in the order Control > WAF > DCEWAF (Fig 1B). However, -Si treatment showed a different pattern with the decreasing growth pattern observed in the following order: DCEWAF > Control > WAF > CEWAF (Fig 1A). The growth response of Day 7 was different from Day 4 with overall higher growth in all groups (Fig 1). Certain combinations of treatments and conditions such as +N+Si in WAF and DCEWAF and–Si in DCEWAF showed significantly higher growth than others (Two-way ANOVA; p < 0.004) (Fig 1B). However, the overall growth of +N+Si and–Si in all the conditions were higher than–N and–N-Si (Two-way ANOVA; p < 0.006) (Fig 1B).

Morphological examination of the cells using an Imaging Flowcytobot revealed interesting features in response to the tested conditions. *P. tricornutum* formed chains of varying lengths in all the conditions (S1 Fig). On day 4, the overall number of cells in chain increased (Two-way ANOVA; p < 0.0001), with +N+Si treatment of each condition showed the least number of cells in chains (S2B Fig). However, this observation was statistically significant only in WAF condition (Two-way ANOVA; p < 0.0007) (S2B Fig). Moreover, CEWAF had the lower number of cells in chains compared to Control and WAF (Two-way ANOVA; p < 0.0002) (S2B Fig). The overall number of cells in chains increased on Day 7 for most treatments and conditions, especially for +N+Si treatment in all conditions (Two-way ANOVA; p < 0.0003) (S2C Fig). Due to an unfortunate incident associated with sample perseveration, these results were not recorded for–N treatments. As we observed large variation in number of cells in chains, no specific trends associated with a given condition or treatment was observed across all three observation time points.

## 3.2 Photo-physiological response of *P. tricornutum*

Photo-physiological responses are reported for Day 1 and 4 to highlight the initial effects and acclimation strategy of *P. tricornutum* in different treatments and conditions. Chlorophyll *a* per cell was higher in +N+Si and–Si treatments compared to–N and–N-Si for Control, WAF and DCEWAF conditions on Day 1, although statistically, only–Si had significantly higher chlorophyll *a* per cell in the DCEWAF compared to–N and–N-Si (Two-way ANOVA; p < 0.002) (Fig 2A). On day 4, a clear pattern of higher chlorophyll *a* per cell in the +N+Si and–Si treatments compared to–N and–N-Si was observed for Control and WAF (Two-way ANOVA; p < 0.02) (Fig 2B). A similar increase was observed in DCEWAF and CEWAF but on nearly half the scale. However, these differences were not significant. Chlorophyll *a* per cell in–N and–N-Si treatments in all the conditions were more or less similar (Fig 2B).

On Day 1, photosynthetic efficiency measured as $F_v/F_m$ was very similar across all the treatments for the Control and WAF, however the values were slightly lower in DCEWAF (Two-way ANOVA; p < 0.002) and also significantly lower in CEWAF (Two-way ANOVA; p < 0.0001) (Fig 3A). On Day 4, the $F_v/F_m$ values were similar between–Si and +N+Si across all the conditions and the same was true for–N and–N-Si (Fig 3B). The $F_v/F_m$ values for–Si and +N+Si were higher than–N and–N-Si treatments for all the conditions except CEWAF, wherein it showed an opposite pattern (Two-way ANOVA; p < 0.002) (Fig 3B). CEWAF also had the lowest $F_v/F_m$ values for–Si and +N+Si compared to the other conditions (Two-way ANOVA; p < 0.0001) (Fig 3B).

Light harvesting ability (α) of PS II was slightly higher in–Si and +N+Si treatments compared to–N and–N-Si for all the conditions on both time points (S3 Fig). However, these differences were not statistically significant for all the conditions. The absorption cross-section area of PS II (σ), a proxy of the size of light harvesting antennae, was similar between–Si and +N+Si in all conditions and significantly higher compared to–N and–N-Si for all the

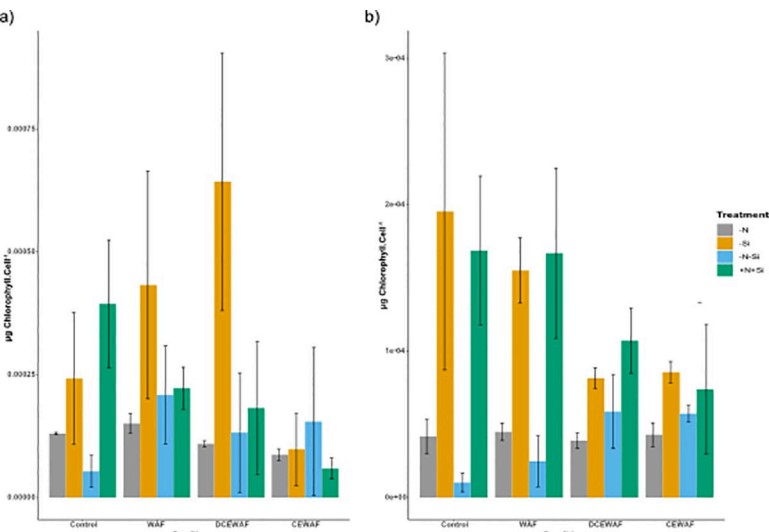

**Fig 2. Pigment concentration of *P. tricornutum*.** a) average chlorophyll *a* per cell on Day 1 (μg. cell$^{-1}$), b) average chlorophyll *a* per cell on Day 4 (μg. cell$^{-1}$) (± standard deviation) under different treatments and conditions (n = 3). The symbols–N, -Si, -N-Si, and +N+Si indicate nitrogen limited, silica limited, both nitrogen and silica limited and nitrogen and silica replete treatments.

conditions on both time points (Two-way ANOVA; p < 0.0001) (Fig 4). Interestingly, the values of σ were slightly higher in–N-Si compared to–N on Day 1; this pattern was significantly reversed on Day 4 (Two-way ANOVA; p < 0.0001) (Fig 4).

The connectivity between PS II (ρ), which measures the ability of the photosystems to redistribute the excitation energy, was significantly higher for–N and–N-Si treatments than–Si and +N+Si in all the conditions on Day 1 (Two-way ANOVA; p < 0.0001). However, these differences diminished on Day 4 with values being similar for all the treatments in all the conditions

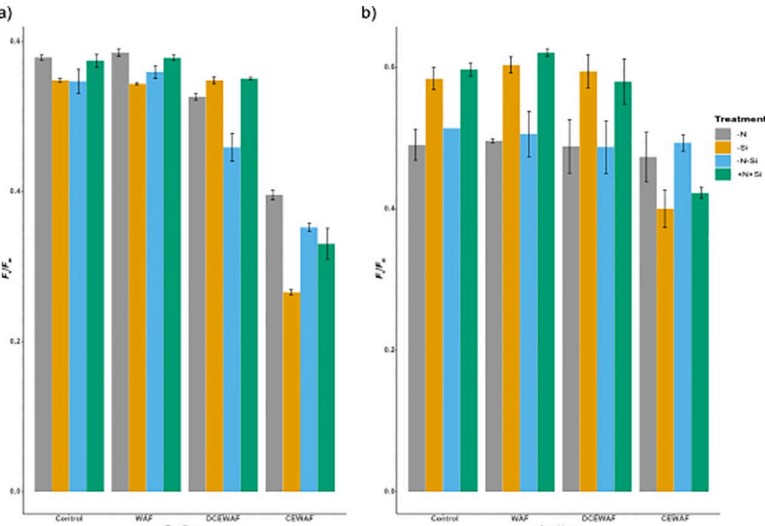

**Fig 3. Photosynthetic efficiency of *P. tricornutum*.** a) average maximum quantum yield on Day 1 ($F_v/F_m$; relative units), b) average maximum quantum yield on Day 4 ($F_v/F_m$; relative units) (± standard deviation) under different treatments and conditions (n = 3). The symbols–N, -Si, -N-Si, and +N+Si indicate nitrogen limited, silica limited, both nitrogen and silica limited and nitrogen and silica replete treatments.

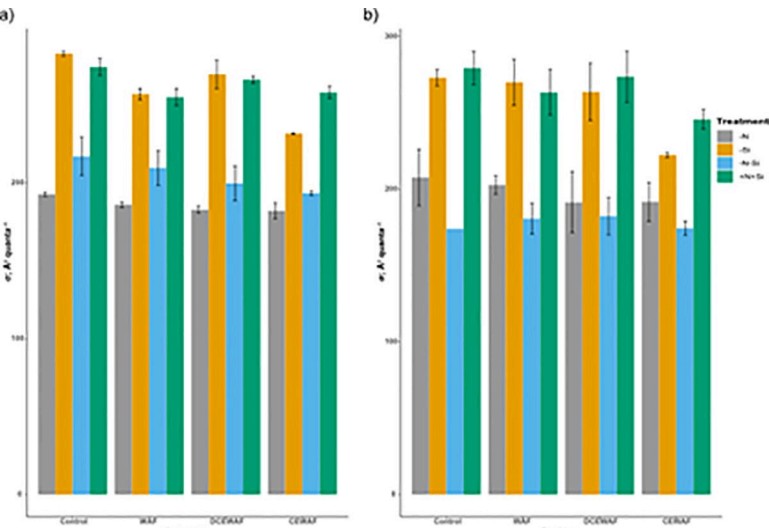

**Fig 4. Photosystem II antennae size of *P. tricornutum*.** a) average absorption cross-section area on Day 1 ($\sigma$; Å$^2$ quanta$^{-1}$), b) average absorption cross-section area on Day 4 ($\sigma$; Å$^2$ quanta$^{-1}$) ($\pm$ standard deviation) under different treatments and conditions (n = 3). The symbols –N, -Si, -N-Si, and +N+Si indicate nitrogen limited, silica limited, both nitrogen and silica limited and nitrogen and silica replete treatments.

(Fig 5A & 5B). The rate of $Q_A$ re-oxidation ($\tau$), which measures the rate at which the primary electron acceptor of $Q_A$ can donate its electron was significantly slower for –N-Si treatment than the rest in all the conditions on Day 1 (Two-way ANOVA; $p < 0.0001$) (Fig 5C). Day 4 $\tau$ values showed similar patterns to Day 1 (Two-way ANOVA; $p < 0.0001$), but the values of –N-Si in the Control treatment was slightly slower than the rest, but not significantly (Fig 5D).

### 3.3 EPS produced by *P. tricornutum*

Due to an unfortunate incident associated with sample perseveration, EPS composition could not be assessed for DCEWAF and CEWAF conditions and hence values are only reported for Control and WAF. The neutral sugars content of the EPS in the Control and WAF conditions was similar between all the treatments (Fig 6A). Protein content of the EPS showed similar patterns to the neutral sugars, except in WAF where the values were much higher for –Si and +N+Si treatments compared to –N and –N-Si; however, these differences were only significant for –Si (Two-way ANOVA; $p < 0.05$) (Fig 6B). Uronic acid content of the EPS was similar across all the treatments in both Control and WAF (Fig 6C). Neutral sugars and uronic acids were summed to estimate carbohydrate content to determine the protein to carbohydrate ratio (P/C) of EPS. P/C was higher for –Si and +N+Si treatments compared to –N and –N-Si; however, due to large variations these differences were not statistically significant (Fig 6D).

### 3.4 Interaction of oil exposure with dispersant and nutrient limitation on growth of *P. tricornutum*

The Control condition (+N+Si) was excluded from this analysis as the cultures were not exposed to oil and/or dispersant. The marginal effects of increasing oil concentrations on relative cellular levels of *P. tricornutum* derived from generalized linear model was overall negative (Two-way ANOVA; $p = 2.973e-06$) (Fig 7A). To test whether this inverse relationship was further enhanced by interaction between presence of dispersants and nutrient limitation, we added the interaction of different treatments and conditions as a factorial variable in the

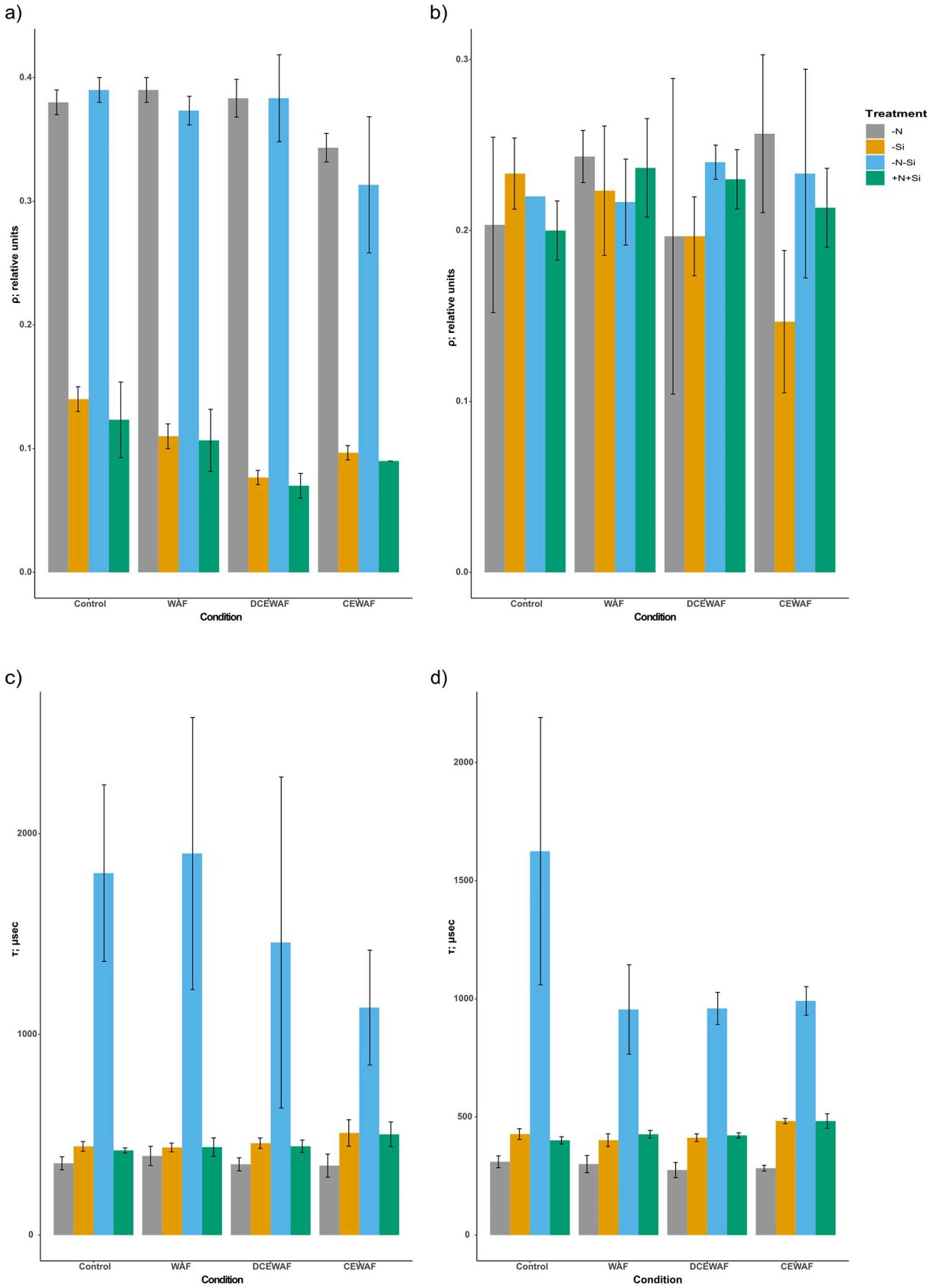

**Fig 5. Photosynthetic physiology of *P. tricornutum*.** a) average connectivity of Photosystem II on Day 1 (ρ; relative units), b) average connectivity of Photosystem II on Day 4 (ρ; relative units), c) average QA re-oxidation rates on Day 1 (τ; μsec), d) average QA re-oxidation rates on Day 4 (τ; μsec) (± standard deviation) under different treatments and conditions (n = 3). The symbols–N, -Si, -N-Si, and +N+Si indicate nitrogen limited, silica limited, both nitrogen and silica limited and nitrogen and silica replete treatments.

generalized linear model. The results suggested significant effects of oil concentration, different conditions, treatments and the interaction between different treatments and conditions (Two-way ANOVA; p < 0.0001). Oil concentration, nutrient limitation and the interaction between nutrient limitation and presence/absence of dispersant showed significant effects in the model (Two-way ANOVA; p < 0.0001). Overall negative effects on relative cellular growth was observed for each factor individually and significant effects were observed for factors such as oil concentration, DCEWAF, and–N with–N-Si and WAF as the reference level (S1 Table). In addition, relative to–N-Si and WAF interaction, positive effects on cellular growth were observed in all the interactions, with significant effects seen in DCEWAF (–N), DCEWAF (–Si), and CEWAF (+N+Si) (S1 Table). We further analyzed the interaction between the treatments and the conditions by comparing the cell-means for the interactions of highest order between factors. Overall, the marginal means of the different conditions were not significant, however, the treatments especially–N, -Si and +N+Si were significantly different (S4 Fig) (One-way ANOVA; p = 0.003). Treatment -N-Si had negative effects on the cell-means of DCEWAF and CEWAF relative to +N+Si; however, these effects were not significant.–N had negative effects on the cell-means in all the conditions, but these effects were only significant in WAF and DCEWAF compared to +N+Si (Chisq Test, p < 0.05) (Fig 7B).–Si had a significantly positive effect on the cell-means of DCEWAF (Chisq Test, p < 0.0001) (Fig 7B). On the other hand, the effects of condition WAF were not significantly different from CEWAF on the cell-means of all the nutrient treatments (Fig 7C). However, the effects of DCEWAF were slightly negative on the cell-means of–N-Si (Chisq Test, p = 0.0563) and significantly positive on the cell-means of–Si relative to CEWAF (Chisq Test, p < 0.0001) (Fig 7C).

## 4. Discussion

Following the Deepwater Horizon oil spill, several studies were conducted to understand the consequences of oil and dispersant exposure on phytoplankton [6, 7, 25–27, 35, 37, 43, 44] given the significant role they play in ecosystems. Several of these studies point to diatoms and dinoflagellates as the most resilient groups to oil exposure [5, 6, 26, 27, 43], with multiple species exhibiting resistance to oil exposure including *P. tricornutum* [26, 45]. Here, we used *P. tricornutum* to study the combined effects of nutrient limitation (N and Si) and oil exposure in the presence and absence of dispersant, as studies have shown that the Gulf of Mexico is likely to be nitrogen and/or silica limited during summer [18–21]. Taken together with the increasing oil exploration in the Gulf of Mexico, an oil spill during a nutrient limited period is quite likely. In addition, we also explored the hypothesis of the role of silica on diatoms when exposed to oil, as it is suggested that the interference of silica transport by oil can lead to negative growth effects [28–31].

*P. tricornutum* exhibited lower growth in both the nitrogen limited treatments (-N, and–N-Si) compared to nitrogen replete treatments (–Si and +N+Si), which suggested a dominant effect of nitrogen on growth. Similar effects of nitrogen limitation on growth have been previously observed for *P. tricornutum* [46, 47] and other phytoplankton species [15, 48]. Given the major role nitrogen plays in the cell, this observation is not unexpected. Unsurprisingly on Day 4, all the nutrient treatments under Control condition showed overall higher growth than in other conditions, except–Si in DCEWAF. On the last day of the experiment, the growth in

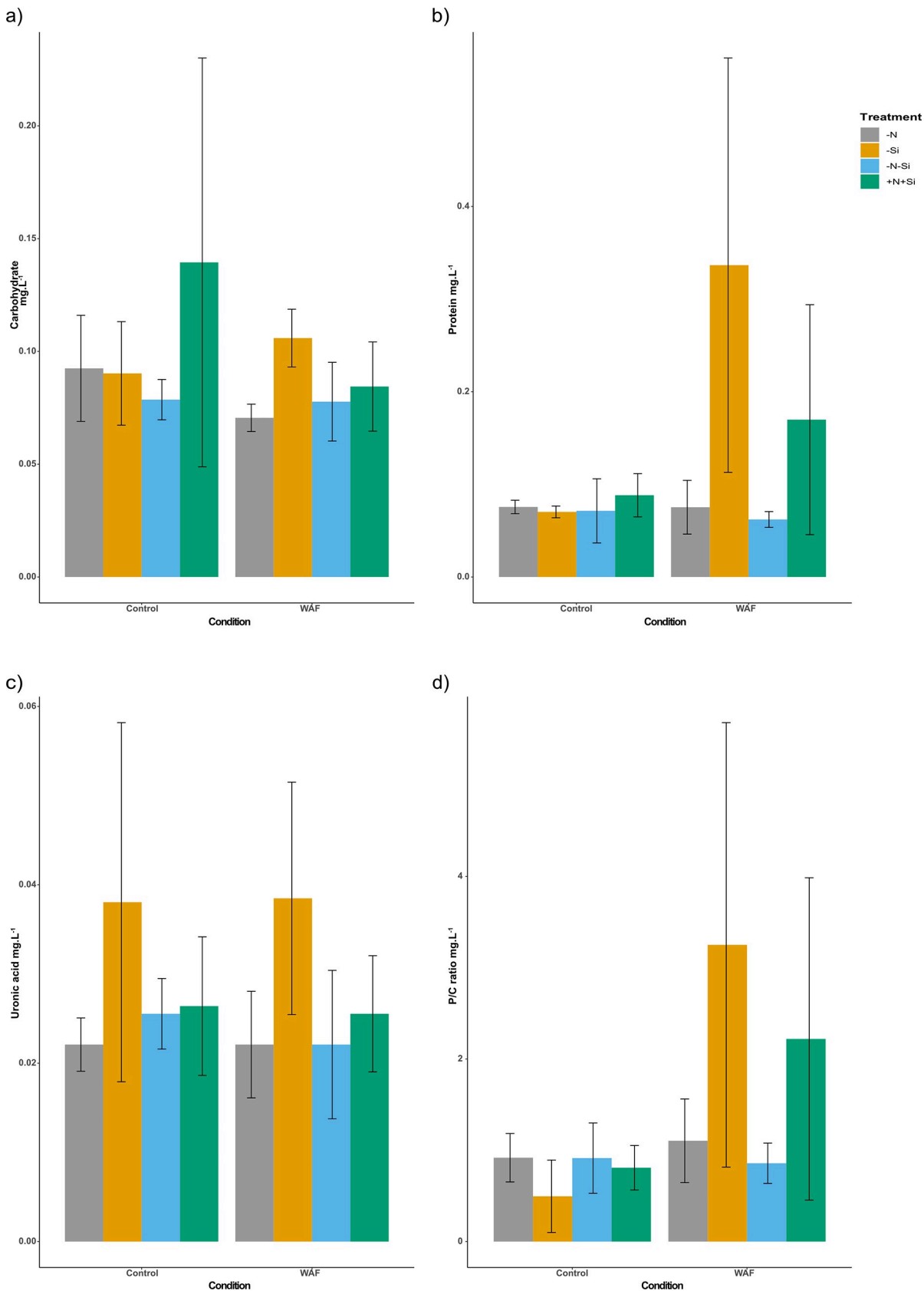

**Fig 6. EPS production by *P. tricornutum*.** a) average carbohydrate content of EPS on Day 4 (mg. L-1), b) average protein content of EPS on Day 4 (mg. L-1), c) average uronic acid content of EPS on Day 4 (mg. L-1), d) average protein to carbohydrate ratio of EPS on Day 4 (mg. L-1) (± standard deviation) under different treatments and conditions (n = 3). The symbols–N, -Si, -N-Si, and +N+Si indicate nitrogen limited, silica limited, both nitrogen and silica limited and nitrogen and silica replete treatments.

the Control condition was actually lower than WAF and DCEWAF, and similar to CEWAF conditions. This suggests a lag in growth phase in WAF and DCEWAF, while the relatively lower increase in growth on Day 7 compared to Day 4 in CEWAF suggests growth inhibition. Such effects of CEWAF have been observed previously [6, 7, 26]. We attribute this effect to nearly 15-fold and 2.7-fold higher concentration of oil than WAF and DCEWAF, respectively. These higher oil concentrations might occur in the presence of dispersant near the site of oil spills; however, the dilution effect of flowing seawater is likely to reduce duration of such high oil concentration pockets.–Si had opposite effects on growth in WAF and DCEWAF. Although the decreased growth in–Si WAF aligns well with the hypothesis of negative impacts of oil on growth due to compromised silica transport [28–31], the increased growth observed in–Si DCEWAF, which had ~5.4-fold higher oil concentration than WAF and the indifference in growth under–Si Control suggest that this might not be true. However, *P. tricornutum* has been previously shown to grow unimpeded under the absence of silica [49]. Therefore, the observed opposite effects of growth in WAF and DCEWAF requires further investigation.

Interestingly, we also observed *P. tricornutum* growing in chains for all the nutrient treatments, including +N+Si Control condition, with the number of cells in the chains increasing with time. Such morphological feature of *P. tricornutum* occurring in chains has been reported in the past [50–53], although the physiological importance and the reasons that causes this morphological form remains to be unknown. It is thought that chain-like morphology can decrease the overall sinking rates of phytoplankton by increasing its viscosity [54, 55]; therefore, more attention needs to be given to the effects of nutrient limitation and pollutants on the morphology of *P. tricornutum* as it may have major implications in the organic matter cycling in the ocean. Our findings show that under the CEWAF condition smaller chains formed with significantly lower number of cells, especially in–Si and +N+Si.

Chlorophyll *a* concentrations per cell were significantly lower for both the nitrogen limited treatments compared to the nitrogen replete. This is not surprising, as nitrogen is an important component for chlorophyll biosynthesis, as glutamate or glycine are required for the production of 5-Aminolevulinic acid, a precursor of chlorophyll biosynthesis [56–58]. Nitrogen limitation leading to decreased cellular chlorophyll levels has been widely recorded for *P. tricornutum* and various phytoplankton species [15, 59–61].

Interestingly, the levels of chlorophyll *a* per cell were lower in DCEWAF and more so in CEWAF, which aligns well with the lower growth observed in these treatments. Maximum photosynthetic efficiency ($F_v/F_m$) was also lower in DCEWAF and significantly lower in CEWAF on Day 1, which could be due to the overall lower chlorophyll *a* content in these conditions and the initial shock of being exposed to higher oil concentrations (13–37 mg.L$^{-1}$) compared to WAF (2.53 mg.L$^{-1}$). However, on Day 4, photosynthetic efficiency in DCEWAF was similar to Control and WAF, despite the lower growth and chlorophyll *a* per cell. On the other hand, the photosynthetic efficiency was higher in–N and–N-Si compared to–Si and +N+Si in CEWAF, a pattern completely opposite from other conditions. These observations suggest treatment and/or condition specific responses and adaptations in *P. tricornutum*. Further analysis of photosynthetic parameters such as absorption cross-section area of PS II ($\sigma$) suggested that the size of the photosynthetic antennae required to harvest light was smaller in both the nitrogen limited treatments on both day 1 and 4 relative to–Si and +N+Si. Chlorophylls are an integral component of photosynthetic antennae [62]; therefore, lower chlorophyll *a* per cell

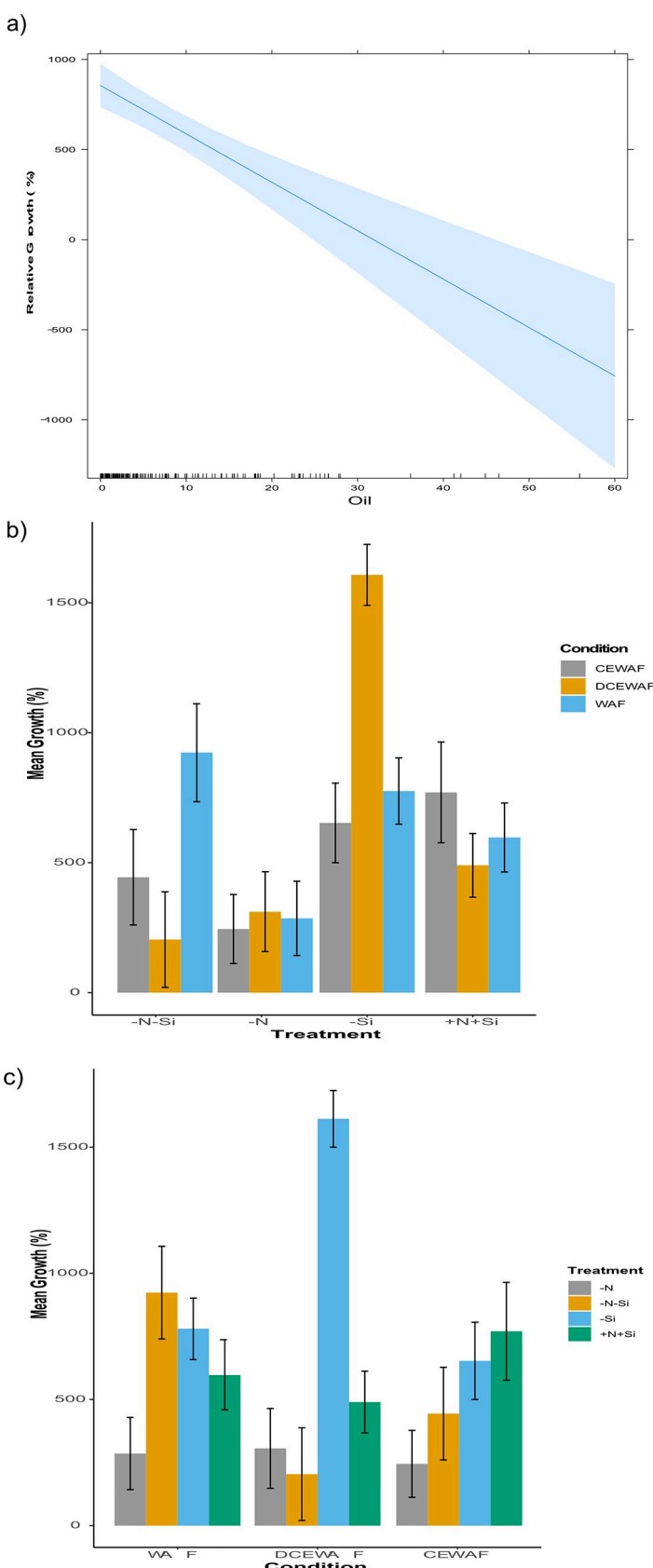

**Fig 7. Generalized linear modelling and interaction effects.** a) Marginal effects of oil concentration on relative growth of *P. tricornutum*, b) Interactive effects of various nutrient treatments on growth of *P. tricornutum* in different conditions, c) Interactive effects of various conditions on growth of *P. tricornutum* in different nutrient treatments. The symbols –N, -Si, -N-Si, and +N+Si indicate nitrogen limited, silica limited, both nitrogen and silica limited and nitrogen and silica replete treatments.

content of nitrogen limited conditions may have negatively affected the photosynthetic antennae size in these treatments. Connectivity between PS II reaction centers ($\rho$) is a measure of the probability that an excitation energy moves from a closed reaction center to another center in any state [63]. The higher $\rho$ in nitrogen limited treatments compared to –Si and +N+Si on Day 1 is not surprising as previous reports have shown that higher connectivity among PS II increases the "effective" antennae size thereby compensating for the smaller size of the PS II antennae [64–66]. Interestingly, there was no significant difference observed between connectivity on Day 4 amongst all the treatments and conditions. These differences in photosynthetic physiology highlight the various changes a cell undergoes from initial shock to 4 days of continuous oil exposure. These physiological adaptations, such as higher connectivity in lower antennae PS II units, may have allowed the nitrogen limited systems to maintain a relatively similar light harvesting ability ($\alpha$) to nitrogen replete treatments. Interestingly, the rate of $Q_A$ re-oxidation ($\tau$) was significantly higher in –N-Si treatment across all the conditions. This indicates that the electron transfer rates between PSII and PSI were higher in –N-Si treatment compared to others suggesting higher rates of light reaction of photosynthesis; however, more research is needed to explain this phenomenon.

EPS production by phytoplankton, a relatively common phenomenon, was reported to be enhanced during Deepwater Horizon oil spill [67], which is consistent with suggestions that EPS can also be a cellular response to stress [68]. We found that the carbohydrate (neutral sugar and uronic acid) content of the EPS secreted by *P. tricornutum* exposed to various treatments and conditions did not vary significantly. The protein content was also similar for all the treatments in Control conditions; however, exposure to WAF significantly enhanced the protein content of EPS in non-nitrogen treatments –Si and +N+Si. It has been shown that the protein to carbohydrate ratio of the EPS can influence the formation of marine snow [69], with higher ratio promoting the process as the material is increasingly sticky. The higher protein to carbohydrate ratio seen in response to WAF in nitrogen replete treatments indicates that oil exposure in our treatments were leading to marine oil snow formation, which aligns well with the observation made by Passow et al. (2012) where excess marine snow was observed following the Deepwater Horizon oil spill [67]. This phenomenon, now termed MOSSFA: marine oil snow sedimentation and flocculent accumulation, accounts for up to 31% of the oil returning to the seafloor [68, 70, 71]. Although the specific nature and functions of proteins released remain unknown (see 69), its excretion is a response to oil exposure. Our observation therefore suggests that an oil spill in coastal zones, which are frequently nutrient limited, could potentially prevent this natural response.

We observed some uniform patterns across our treatments and conditions such as relatively lower growth, chlorophyll *a*, and $\sigma$ in nitrogen limited treatments compared to non-nitrogen limited and initial lower $F_v/F_m$, lower growth, and chlorophyll *a* in the dispersant conditions (CEWAF and DCEWAF). However, various observations differed from these patterns, such as a) initial higher $\rho$ in all nitrogen limited treatments followed by similar values across on Day 4, b) initial lower $F_v/F_m$ in DCEWAF compared to Control followed by similar values on Day 4 despite lower chlorophyll *a* content, and lastly c), higher $\tau$ values across all the –N-Si treatments. These deviations from the general pattern indicate interaction between the treatment and conditions leading to specific acclimations in *P. tricornutum*. We looked for these

interactions by comparing the cellular growth response to oil concentrations measured in our experiment and using our different treatments and conditions as factors in a generalized linear regression model. An important note in the interpretation of this analysis is that the Control condition was excluded, and therefore any positive or negative effects noted are not relative to any treatments of Control. Growth decreased with increasing oil concentration, and factors such as DCEWAF and–N showed significant effects compared to the other factors (-Si, -N-Si, +N+Si, WAF and CEWAF). However, interaction analysis revealed that both nitrogen limited treatments had negative effects, with significant effects seen for–N compared to–N-Si, -Si, and +N+Si in WAF and DCEWAF. This suggests that nitrogen limitation can severely affect the growth of *P. tricornutum* when exposed to oil and a slightly higher concentration of oil resulting from the presence of dispersant (this condition is more likely to occur in a low mixing water zone). We observed lower chlorophyll levels in WAF and DCEWAF under–N, and therefore a lower σ and τ resulting in lower maximum photosynthetic efficiency, which in turn might have affected the growth, as discussed above. Moreover, lack of ability to produce EPS with higher protein content under–N may have reduced any protective roles these proteins may have offered [68]. On the other hand, DCEWAF had a negative effect on–N-Si and a positive effect on–Si compared to +N+Si. This observation is interesting as DCEWAF had opposite effects in the absence of silica depending on nitrogen availability. Although the positive effects of DCEWAF on–Si remains difficult to interpret given our current knowledge, most of the negative effects of DCEWAF on–N-Si can be explained by the relatively lower chlorophyll levels, σ, τ and lower maximum photosynthetic efficiency and proteins in EPS. While one would expect severe effects of CEWAF compared to WAF and DCEWAF based on previous studies [6, 25, 26], the relatively reduced growth observed in CEWAF dampens any response and interaction with nutrient treatments.

Several studies have shown that oil and dispersant exposure can restructure the phytoplankton community composition favoring diatoms and dinoflagellates, however our study shows that this rule comes with an asterisk. Our study, with one of the oil and dispersant resistant diatoms *P. tricornutum*, shows that environmental conditions such as nutrient concentrations and ratios, has to be taken into consideration during an oil spill. Potential nitrogen and/or silica limitation can favor dinoflagellates over diatoms, thereby altering several biogeochemical processes. This finding is further emphasized with the ability of dinoflagellates to grow in low nutrient conditions [72, 73]. Recent studies performed as a result of the Deepwater Horizon oil spill also reveal that dinoflagellates often do well after an oil spill, and in some cases, even causing harmful algal bloom events [74, 75]. Therefore, we hypothesize that potential nitrogen and/or silica limitation can favor dinoflagellates over diatoms, thereby altering several biogeochemical processes. Further studies into investigating the effects of nutrient limitations and oil and dispersant exposure would be beneficial. Another example include the difference in sedimentation pattern of diatoms versus dinoflagellates, with diatoms sinking faster than dinoflagellates that tend sink in cyst form or lyse in the water column potentially slowing down vertical transport of organic matter [76]. Such a shift is also likely to alter the fate of MOSSFA phenomenon during an oil spill, which is one of the major processes deciding the fate of the oil [68, 70, 71].

## 5. Conclusion

Nutrient limitation events are common in the ocean and are prevalent in the Gulf of Mexico [18–20]. With increasing oil exploration activities in the Gulf of Mexico, it is only a matter of time before the next oil spill. Moreover, the likelihood of an oil spill with nutrient limitation is high; therefore, exploring the combined effects of oil spills and nutrient limitation is vital.

Here, we demonstrate how oil spills and the use of dispersants can affect one of the most oil resistant diatoms when such disasters coincide with nutrient limitation. We show that overall nitrogen limitation can severely stunt the resistant nature of *P. tricornutum* to oil exposure. Also, the hypothesis of negative impacts of oil on growth due to compromised silica transport [28–31] may not be true; however, more research is needed. Additionally, we show significant interactive effects of oil exposure with nutrient limitation and dispersants that could potentially shift the phytoplankton community structure towards dinoflagellate during an oil spill.

## Supporting information

**S1 Fig. Raw growth response of *P. tricornutum* under different treatments and conditions (n = 3) on days 1, 4 and 7.** The symbols–N, -Si, -N-Si, and +N+Si indicate nitrogen limited, silica limited, both nitrogen and silica limited and nitrogen and silica replete treatments.
(TIFF)

**S2 Fig. Morphological changes in *P. tricornutum*.** a) average number of cells in chains on Day 1, b) average number of cells in chains on Day 4, c) average number of cells in chains on Day 7 (± standard deviation) under different treatments and conditions (n = 3). The symbols–N, -Si, -N-Si, and +N+Si indicate nitrogen limited, silica limited, both nitrogen and silica limited and nitrogen and silica replete treatments.
(TIFF)

**S3 Fig. Photosynthetic physiology of *P. tricornutum*.** a) average light harvesting ability on Day 1 ($\alpha$; µmol e⁻ µmol photons), b) average light harvesting ability on Day 4 ($\alpha$; µmol e⁻ µmol photons) (± standard deviation) under different treatments and conditions (n = 3). The symbols–N, -Si, -N-Si, and +N+Si indicate nitrogen limited, silica limited, both nitrogen and silica limited and nitrogen and silica replete treatments.
(TIFF)

**S4 Fig. Generalized linear modelling and interaction effects.** a) Marginal effects of different conditions on relative growth of *P. tricornutum*, b) Marginal effects of different nutrient treatments on relative growth of *P. tricornutum*. The symbols–N, -Si, -N-Si, and +N+Si indicate nitrogen limited, silica limited, both nitrogen and silica limited and nitrogen and silica replete treatments.
(TIFF)

**S1 Table. Summary of generalized linear modelling of relative growth vs oil concentration with interaction of different treatments and conditions as factorial variables.**
(DOCX)

## Acknowledgments

We would like to acknowledge Brittany light for her help in the lab work.

## Author Contributions

**Conceptualization:** Manoj Kamalanathan, Antonietta Quigg.

**Data curation:** Manoj Kamalanathan, Jessica Hillhouse, Noah Claflin, Talia Rodkey, Andrew Mondragon, Alexandra Prouse, Michelle Nguyen.

**Formal analysis:** Manoj Kamalanathan.

**Funding acquisition:** Antonietta Quigg.

**Methodology:** Jessica Hillhouse, Noah Claflin, Talia Rodkey, Andrew Mondragon, Alexandra Prouse, Michelle Nguyen.

**Project administration:** Antonietta Quigg.

**Software:** Manoj Kamalanathan.

**Supervision:** Antonietta Quigg.

**Validation:** Manoj Kamalanathan.

**Visualization:** Manoj Kamalanathan.

**Writing – original draft:** Manoj Kamalanathan.

**Writing – review & editing:** Manoj Kamalanathan, Jessica Hillhouse, Noah Claflin, Talia Rodkey, Alexandra Prouse, Antonietta Quigg.

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
