## [Decision Letter · Decision Letter 0]

27 Jul 2021

PONE-D-21-04883

Influence of nutrient status on the response of the diatom Phaeodactylumtricornutum to oil and dispersant

PLOS ONE

Dear Dr. Kamalanathan,

Thank you for submitting your manuscript to PLOS ONE. After careful consideration, we feel that it has merit but does not fully meet PLOS ONE’s publication criteria as it currently stands. Therefore, we invite you to submit a revised version of the manuscript that addresses the points raised during the review process.

Please address all the comments by reviewers, particularly those requesting other experiments or monitoring effort.

We look forward to receiving your revised manuscript.

Kind regards,

Andrea Franzetti

Academic Editor

PLOS ONE

Journal Requirements:

 AQ

ADDOMEX2

Gulf of Mexico Research Initiative

https://gulfresearchinitiative.org/

NO

6. Please upload a new copy of Figures 2,3 and 4 as the detail is not clear. Please follow the link for more information: https://blogs.plos.org/plos/2019/06/looking-good-tips-for-creating-your-plos-figures-graphics/" https://blogs.plos.org/plos/2019/06/looking-good-tips-for-creating-your-plos-figures-graphics/.

Reviewers' comments:

Reviewer's Responses to Questions

**Comments to the Author**

1. Is the manuscript technically sound, and do the data support the conclusions?

Reviewer #1: No

Reviewer #2: Partly

2. Has the statistical analysis been performed appropriately and rigorously? 

Reviewer #1: No

Reviewer #2: Yes

3. Have the authors made all data underlying the findings in their manuscript fully available?

Reviewer #1: No

Reviewer #2: Yes

4. Is the manuscript presented in an intelligible fashion and written in standard English?

Reviewer #1: Yes

Reviewer #2: Yes

5. Review Comments to the Author

Reviewer #1: Review for PLOS ONE

The authors present a study on the cumulative impacts of oil and nutrient limitation, which would be a interest to algal physiologists. The experimental design is ambitious and includes numerous algal endpoints, although many are missing data. Unfortunately, the experiment was not conducted effectively.

Oil is not uniformly soluble in water, so unfortunately, as the authors did not measure the concentrations of oil in water, the dose used can not be determined. The reader knows that the phytoplankton physiology was changed, but not by what. As a consequence, the paper must be rejected. (You have a y-axis, but your x-axis is effectively blank). We have no way of knowing whether the concentrations of oil used are environmentally realistic, or how they compare to other studies reported in the literature. Estimated Oil Equivalents can not be converted to a PAH or TRH concentration. Without having dose quantified, your comparisons between WAF; CEWAF, etc. are meaningless.

The authors are referred to recent reviews by Peter Hodson for additional information and for details and points to consider in future studies.

In addition, the authors discuss the impacts of nutrient limitation in comparisons where the nutrient abundant treatments were missing. This is inappropriate.

The remainder of my comments are to help guide the authors in the preparation of other manuscripts

Why was only one dose of each treatment utilised? It would be better to have a series of doses to determine thresholds

10^5 cells per ml is quite high for nutrient limited concentrations. Wouldn’t 10^4 be more realistic?

Many endpoints are incomplete due to “technical issues” – why is the endPoint included at all if the experiment can not be repeated? It’s especially challenging to interpret the results when the nutrient replete

The discussion is largely repetitive of the results. Please try to put your work in more context.

The sentence that “With increasing oil exploration activities in the Gulf of Mexico, it is only a matter of time before the next oil spill” is repeated several times in the paper. Please paraphrase yourself.

The writing is frequently sensational- how is the interaction “remarkable” for example?

Table 1 would be better as supplemental material

Figures

Figure 1 – it would be helpful to mark which growth rates are significantly different with an asterisk or similar. It’s very difficult to follow the description in the text.

Figure 2-6 – again, mark those the are significantly different, treatment would be a better x axis than condition

Figure 7a – the uncertainty in the growth relationship negates the values of this graph, and one wonders how it was derived given that only one treatment was used?

For the others, why are lines drawn between treatments? These are not continuous variables

Reviewer #2: Kamalanathan et al. have examined influence of nutrient status (N and Si) on the response of the diatom Phaeodactylum tricornutum to oil and dispersant. The experimental methods used in this study seem to be appropriate, and the manuscript is generally well written. However, there are major and minor concerns in the manuscript.

[Major]

#01. In this study, growth response of P. tricornutum depending on chemical status (nutrients and oil) was analyzed throughout comparing relative abundances (at Day 4 and 7) of P. tricornutum in control and treatments. I am wondering how authors determine that the dates (sample collection date, Day 4 and 7) are appropriate for this study? Is there any specific reason/criterion (e.g considering growth phase of P. tricornutum or this diatom reached the maximum cell density at day 7??)? If so, the description regarding this should be added to material and methods.

I would like to recommend that authors reconsider the use of “relative growth” in this study. In my thought, “growth curve” graph (variation in cell density depending on time) would be more suitable for this study. This graph should be provided as supplementary data even if authors think the current format is more suitable. Besides, how do authors calculate this relative growth? For readers, the equation should be described in material and methods.

#02. To reach a robust conclusion, it is highly necessary to measure the concentration of nutrients (N and Si, inorganic+organic form) in samples. As you might know, many kinds of chemicals are present in crude oil. Thus, in my thought, there are possibility that nutrient status can be changed in oil treatments (WAF, CEWAF….), if the diatom culture which were used in this study is not axenic; bacterial communities can affect variation in chemical status when they are exposed to oil, since they can degrade oil into various form (chemically), and bacteria can also change nutrient bioavailability. If so, this might cause misleading results and/or misinterpretation of this work. In addition, in order to determine whether or not the hypothesis of negative impacts of oil on diatom growth due to compromised silica transport is true, the data (nutrient concentration) should be provided. Besides, I am not sure this diatom can be the best species to test this hypothesis. For example (Fig. 1), the growth of this diatom did not seem to affect Silicate; relative growth was similar regardless of concentration of Silicate..

[Minor]

- Addresses of authors should be corrected.

- L69. In order to reduce confusion, please change “on phytoplankton” to “on the growth of phytoplankton”.

- L89. Change “diatomhas” to “diatom has”.

- L321-323. Discussion on this sentence is thought to be necessary. What causes variation in chain formation/length depending on chemical status (nutrients and oil)

- L411-413 “…..can favor dinoflagellates over diatoms, thereby……”; If there are no data on growth response of dinoflagellates to chemical status (nutrients and oil), this discussion should be more careful. Additionally, only one culture of single diatom species was used in this work.

- L421 “… contributor globally []…” In my guess, the references are missed.

- L420-423 I couldn’t understand why this sentence is described here. If this sentence is necessary, this sentence should be revised for readers. Additionally, it would be more appropriate to add references which are published in more recent.

- Figure caption. For readers, please add the description on each treatment. Additionally, I cannot find description on +N+Si and +Si treatments in materials and methods.

6. PLOS authors have the option to publish the peer review history of their article (what does this mean?). If published, this will include your full peer review and any attached files.

Reviewer #1: No

Reviewer #2: No

---

## [Author Response · Author response to Decision Letter 0]

19 Sep 2021

Response to reviewer’s comments:

Response: We thank the editor and both the reviewers for the insightful comments and the opportunity to address them and submit the revised version. Addressing the comments has helped us make the manuscript scientifically stronger. And we believe we were have addressed them all the concerns raised by the reviewers to our best capacity, we hope they are satisfactory.

Reviewer #1: Review for PLOS ONE

The authors present a study on the cumulative impacts of oil and nutrient limitation, which would be a interest to algal physiologists. The experimental design is ambitious and includes numerous algal endpoints, although many are missing data. Unfortunately, the experiment was not conducted effectively.

Oil is not uniformly soluble in water, so unfortunately, as the authors did not measure the concentrations of oil in water, the dose used can not be determined. The reader knows that the phytoplankton physiology was changed, but not by what. As a consequence, the paper must be rejected. (You have a y-axis, but your x-axis is effectively blank). We have no way of knowing whether the concentrations of oil used are environmentally realistic, or how they compare to other studies reported in the literature. Estimated Oil Equivalents can not be converted to a PAH or TRH concentration. Without having dose quantified, your comparisons between WAF; CEWAF, etc. are meaningless.

Response: We added oil at a concentration of 400μl/L of sea water to make all the WAF, CEWAF and DCEWAF using the method described in The Chemical Response to Oil Spills: Ecological Research Forum (CROSERF). This is a standard method that has been used throughout the oil spill toxicity studies, please see: Ozhan and Bargu, 2014; Faksness et al., 2015; Cohen et al., 2015. Moreover, these papers are also some of the studies cited in the Peter Hodson’s review articles (as per the reviewer’s suggestion below). 

The resulted oil concentrations after using CROSERF method in WAF, DCEWAF and CEWAF was 2.53 mg/L, 13.76 mg/L, and 37.16 mg/L. These concentrations were environmentally realistic and comparable to the total petroleum hydrocarbon concentrations in samples collected near the wellhead at the surface seawater (where phytoplankton are more likely to be found) after DwH incident, and far lower than the average TPH concentration (202.206 mg/L) in the seawater samples analyzed in Sammarco et al. (2013).

We agree with the reviewer about the nature of dissolution of oil in water, however, we did perform oil measurements using Estimated Oil Equivalence method (Wade et al., 2011 & 2017). Even though estimated oil equivalents cannot be directly converted to PAH concentration, it is well documented that they both are strongly correlated. Please see: Wade, T.L., Sweet, S.T., Sericano, J.L., Guinasso, N.L., Diercks, A.R., Highsmith, R.C., Asper, V.L., Joung, D., Shiller, A.M., Lohrenz, S.E. and Joye, S.B., 2011. 

Analyses of water samples from the Deepwater Horizon oil spill: Documentation of the subsurface plume. Monitoring and Modeling the deepwater horizon oil spill: a record-breaking enterprise, 195, pp.77-82. 

Moreover, the estimated oil equivalent values reported in this manuscript are determined from the calibration curve of different concentrations (ranging from 100 to 5000 μg/L) of oil, rather than just relative fluorescence. Therefore we are confident that the oil concentration used in this study is not only environmentally realistic but also comparable to the other studies.

The authors are referred to recent reviews by Peter Hodson for additional information and for details and points to consider in future studies.

Response: Thank you for the recommendation! Our study especially under the context of the oil concentration used follow the same protocols as the papers cited in Peter Hodson’s review articles and some of his own articles (Beyer et al., 2016; Martin et al., 2014). Although, we are aware that there are limitations associated with CROSERF approach, and that there is a dire need to improve the oil in water dispersions, we would request the editor and the reviewer to recognize that it is beyond the scope and not the objective of this study. 

In addition, the authors discuss the impacts of nutrient limitation in comparisons where the nutrient abundant treatments were missing. This is inappropriate.

Response: We fully agree with the reviewer’s comment, we thank the reviewer for pointing it out and we apologize for the mistake! We noticed that this error was only present in the section where we discussed the interaction analysis. We have now addressed this concern by adding appropriate comparison reference (+N+Si and/or –N and/or -Si) to the sentences where it was missing. Please see line no: 397-400, 407-408.

The remainder of my comments are to help guide the authors in the preparation of other manuscripts

Why was only one dose of each treatment utilised? It would be better to have a series of doses to determine thresholds

Response: We agree with the reviewer’s comment, however, we had a total of 16 treatments in triplicates which equals to 48 samples for every analysis we conducted. Certain parameters such as photo-physiological measurements had to be measured right away with fresh samples and takes 15 mins per sample. Having even as few as three different doses would have bought the sample numbers to a total of 144, making determination of such photo-physiogical parameters impossible within the 24hrs period of the day. Therefore, although performing a series of doses although ideal was beyond the scope of this study.

10^5 cells per ml is quite high for nutrient limited concentrations. Wouldn’t 10^4 be more realistic?

Response: We respectfully disagree with the reviewer here, 10^5 cells per ml is a standard concentration of culture used throughout any laboratory phytoplankton studies. 

Many endpoints are incomplete due to “technical issues” – why is the endPoint included at all if the experiment can not be repeated? It’s especially challenging to interpret the results when the nutrient replete

Response: We apologize for this issue! Certain samples were not good due to an unfortunate incident associated with sample preservation and resulted in contamination. However, we would like to point out that there are only two analysis including measurement of cell numbers in chain morphology, and EPS measurement where this happened, all the other data presented in the manuscript are complete.

The discussion is largely repetitive of the results. Please try to put your work in more context.

Response: We respectfully disagree with the reviewer here! The discussion of this paper is divided into 7 sections: A general paragraph outlining the importance of the study, a paragraph discussing the observed growth effects, morphological effects, photo-physiological effects, changes in EPS production, interactive factors with explanation derived from the above mentioned results, and lastly a summary paragraph with a big picture context. Overall, we discussed each aspect of the results with previous observations in the literature and finished with a summary of how it fits in an oil spill context. Therefore, we do not agree that the discussion is largely repetitive.

The sentence that “With increasing oil exploration activities in the Gulf of Mexico, it is only a matter of time before the next oil spill” is repeated several times in the paper. Please paraphrase yourself.

Response: We agree with the reviewer, this sentence was repeated exactly twice in the manuscript and we have now paraphrased the second time it appears. Please see line no: 296-297.

The writing is frequently sensational- how is the interaction “remarkable” for example?

Response: We agree with the reviewer, we apologize for the mistake. We have now replaced the word “remarkable” with “significant”. Please see line no: 442.

Table 1 would be better as supplemental material

Response: We agree with the reviewer. We have now moved Table 1 to supplementary material.

Figures

Figure 1 – it would be helpful to mark which growth rates are significantly different with an asterisk or similar. It’s very difficult to follow the description in the text.

Response: We agree with the reviewer. We have now marked all the significant results with an asterisk.

Figure 2-6 – again, mark those the are significantly different, treatment would be a better x axis than condition

Response: We agree with the reviewer. We did try putting upside down brackets between bars and mark the ones that are significant, but due to multiple treatments, conditions and days in the bar chart, this quickly got very complicated to look at. Hence, we opted out of it. 

Figure 7a – the uncertainty in the growth relationship negates the values of this graph, and one wonders how it was derived given that only one treatment was used?

Response: We respectfully disagree partly with the reviewer here, even after accounting for uncertainty, the data shows decrease from 750 to minus 250%, which highlights a clear trend in the observation. However, we agree with the reviewer on the second half part of the comment. Regarding how the values were derived, we did not provide adequate information in the methods section, which we have addressed in the revised version of the manuscript. Please see line no: 161-169, which states “The effect of oil concentrations on relative cellular levels was analyzed using a generalized linear model in R. The cellular concentrations were normalized for all the experiments by calculating the percent change in growth relative to Day 1 for this analysis. EOE values measured across different time points during the experiments across the different treatments and conditions were used as oil concentration for this analysis.” 

For the others, why are lines drawn between treatments? These are not continuous variables

Response: We agree with the reviewer. We have now changed the line plots to bar plots.

Reviewer #2: Kamalanathan et al. have examined influence of nutrient status (N and Si) on the response of the diatom Phaeodactylum tricornutum to oil and dispersant. The experimental methods used in this study seem to be appropriate, and the manuscript is generally well written. However, there are major and minor concerns in the manuscript.

[Major]

#01. In this study, growth response of P. tricornutum depending on chemical status (nutrients and oil) was analyzed throughout comparing relative abundances (at Day 4 and 7) of P. tricornutum in control and treatments. I am wondering how authors determine that the dates (sample collection date, Day 4 and 7) are appropriate for this study? Is there any specific reason/criterion (e.g considering growth phase of P. tricornutum or this diatom reached the maximum cell density at day 7??)? If so, the description regarding this should be added to material and methods.

Response: We agree with reviewer’s concern here. For all the parameters tested in the experiments, the sampling time points were chosen based on the typical growth curves of P. tricornutum to accommodate the initial time point (Day 1), the logarithmic phase (Day 4) and the stationary phase (Day 7) effects. We have now updated the materials and methods to state the same in line no: 133-136.

I would like to recommend that authors reconsider the use of “relative growth” in this study. In my thought, “growth curve” graph (variation in cell density depending on time) would be more suitable for this study. This graph should be provided as supplementary data even if authors think the current format is more suitable. Besides, how do authors calculate this relative growth? For readers, the equation should be described in material and methods.

Response: We agree with the reviewers concern here, however, due to the large variation in the response of the growth of P. tricornutum to the various conditions and treatments used in this study, it made the graph difficult to read. In order to facilitate easy interpretation of the effects measured in terms of growth inhibition caused under the various conditions and treatments we decided that expressing the data as relative growth would be more appropriate. However, we fully agree that the readers should have the raw growth curve data as well, hence as per reviewer’s suggestion we have included a new supplementary file that has the raw growth curves of P. tricornutum to all the conditions and treatments. Please see supplementary figure 1 and line no: 180-181.

#02. To reach a robust conclusion, it is highly necessary to measure the concentration of nutrients (N and Si, inorganic+organic form) in samples. As you might know, many kinds of chemicals are present in crude oil. Thus, in my thought, there are possibility that nutrient status can be changed in oil treatments (WAF, CEWAF….), if the diatom culture which were used in this study is not axenic; bacterial communities can affect variation in chemical status when they are exposed to oil, since they can degrade oil into various form (chemically), and bacteria can also change nutrient bioavailability. If so, this might cause misleading results and/or misinterpretation of this work. In addition, in order to determine whether or not the hypothesis of negative impacts of oil on diatom growth due to compromised silica transport is true, the data (nutrient concentration) should be provided. Besides, I am not sure this diatom can be the best species to test this hypothesis. For example (Fig. 1), the growth of this diatom did not seem to affect Silicate; relative growth was similar regardless of concentration of Silicate..

Response: We agree with the reviewer’s concern here! Unfortunately, the nutrient concentrations in the samples were not determined, however, crude oil tend to contain less than 0.1-2% of nitrogen (Overton et al., 2016) and 0% silica. Given the minute concentration of oil used in this study with 2 to 37 ppm in WAF to CEWAF, the amount of nitrogen derived from bacterial activity would be negligible compare to the ¼ of the amount of original N in the ASW medium. Therefore, the effect on nitrogen bioavailability caused by nitrogen derivation from crude oil by bacterial activity should be negligible. 

Regarding the comment made by the reviewer on the hypothesis of silica transport, we fully agree with the reviewers comment. However, we have already addressed this concern in the discussion of the manuscript. For example, in line no: 319-320. We state “However, P. tricornutum has been previously shown to grow unimpeded under the absence of silica [49]. Therefore, the observed opposite effects of growth in WAF and DCEWAF requires further investigation.”

[Minor]

- Addresses of authors should be corrected.

- L69. In order to reduce confusion, please change “on phytoplankton” to “on the growth of phytoplankton”.

Response: We apologize for the mistake and have now corrected the sentence as per reviewer’s suggestion. Please see line no: 69-70.

- L89. Change “diatomhas” to “diatom has”.

Response: We apologize for the mistake and have now corrected the sentence as per reviewer’s suggestion. Please see line no: 89.

- L321-323. Discussion on this sentence is thought to be necessary. What causes variation in chain formation/length depending on chemical status (nutrients and oil)

Response: We agree with the reviewer’s concern here. Borowitzka et al., (1977) is one of the few studies that focuses on the chain like morphology of P. tricornutum, however, the study concludes with emphasis on more research needed to understand the reasons that causes such change in morphology. Therefore, we have modified the sentence to state “Such morphological feature of P. tricornutum occurring in chains has been reported in the past [50-53], although the physiological importance and the reasons that causes this morphological form remains to be unknown.” Please see line no: 324-326.

- L411-413 “…..can favor dinoflagellates over diatoms, thereby……”; If there are no data on growth response of dinoflagellates to chemical status (nutrients and oil), this discussion should be more careful. Additionally, only one culture of single diatom species was used in this work.

Response: We agree with the reviewer’s concern here and have modified the sentence to state “Therefore, we hypothesize that potential nitrogen and/or silica limitation can favor dinoflagellates over diatoms, thereby altering several biogeochemical processes. Further studies into investigating the effects of nutrient limitations and oil and dispersant exposure would be beneficial.” Please see line no: 425-428.

- L421 “… contributor globally []…” In my guess, the references are missed.

Response: We apologize for the mistake here, and having considered the comment below, we have deleted the sentence.

- L420-423 I couldn’t understand why this sentence is described here. If this sentence is necessary, this sentence should be revised for readers. Additionally, it would be more appropriate to add references which are published in more recent.

Response: We agree with reviewer’s concern here that the sentence feels out of context for this study and have deleted them in the revised version of the manuscript.

- Figure caption. For readers, please add the description on each treatment. Additionally, I cannot find description on +N+Si and +Si treatments in materials and methods.

Response: We apologise for the mistake! We have added the following caption for all the figures now “The symbols –N, -Si, -N-Si, and +N+Si indicate nitrogen limited, silica limited, both nitrogen and silica limited and nitrogen and silica replete treatments.” We have also now provided the description of +N+Si and +Si treatments in the methods. Please see line no: 120-121.

---

## [Editor Report · Decision Letter 1]

21 Oct 2021

Influence of nutrient status on the response of the diatom Phaeodactylumtricornutum to oil and dispersant

PONE-D-21-04883R1

Dear Dr. Kamalanathan,

We’re pleased to inform you that your manuscript has been judged scientifically suitable for publication and will be formally accepted for publication once it meets all outstanding technical requirements.

Kind regards,

Andrea Franzetti

Academic Editor

PLOS ONE
---

## [Editor Report · Acceptance letter]

3 Nov 2021

PONE-D-21-04883R1 

Influence of nutrient status on the response of the diatom *Phaeodactylum tricornutum* to oil and dispersant 

Dear Dr. Kamalanathan:

I'm pleased to inform you that your manuscript has been deemed suitable for publication in PLOS ONE. Congratulations! Your manuscript is now with our production department. 

Kind regards, 

on behalf of

Dr. Andrea Franzetti 

Academic Editor

PLOS ONE